# OpenReview forum: "Surgery: Mitigating Harmful  Fine-Tuning for Large Language Models via Attention Sink"
_ICML.cc/2026/Conference — ICML 2026 spotlight_

### Official Review · Reviewer_dUh7 · 2026-03-02

**Soundness:** 3
**Presentation:** 3
**Significance:** 3
**Originality:** 3
**Overall Recommendation:** 5
**Confidence:** 5

**Summary:**

The paper addresses the vulnerability of safety-aligned Large Language Models (LLMs) to Harmful Fine-Tuning (HFT), where a small amount of malicious data during customization can compromise a model's safety guardrails. The authors investigate the role of attention sinks—initial tokens that disproportionately attract attention—and introduce a new metric called sink divergence. This metric measures the difference in attention sink values when a model processes harmful versus safe (refusal) responses.Key findings include the separable sink divergence hypothesis, which states that attention heads associated with learning harmful patterns can be identified by their positive sink divergence. Based on this, the authors propose Surgery, a defense mechanism that adds a sink divergence suppression regularizer during fine-tuning. This regularizer steers attention heads toward a negative sink divergence state, effectively preventing the model from learning harmful behaviors without significantly degrading performance on benign tasks.

**Compliance With Llm Reviewing Policy:**

Affirmed.

**Final Justification:**

Thanks for the response. I have raised the score to 5.

**Key Questions For Authors:**

How does this method differ from direct full-parameter gradient surgery [1,2], even though they seem to be the same in terms of both effect and principle?

[1] Yi, B. et. al. Gradient surgery for safe llm fine-tuning. arXiv preprint arXiv:2508.07172.

[2] Zhang, J. et. al. Understanding and Preserving Safety in Fine-Tuned LLMs. arXiv preprint arXiv:2601.10141.

**Limitations:**

Yes

**Strengths And Weaknesses:**

Strengths:

* Novel Perspective: This is the first work to link the well-known "attention sink" phenomenon to the safety behaviors of LLMs during fine-tuning.

* Mechanistic Foundation: The "separable sink divergence hypothesis" provides a clear, actionable theory for why certain parts of the model become "harmful" and how to target them.

Weaknesses:

* Dependency on Dataset Pairs: The method assumes the service provider has access to both a "harmful" dataset (harmful prompt + harmful answer) and an "alignment" dataset (harmful prompt + safe answer) to calculate initial sink divergence. This data may not always be readily available or perfectly matched for specialized domains.

* Hyperparameter Sensitivity: The defense relies on a regularization coefficient ($\lambda$). While the paper explores different values, there is some variance in performance and fine-tuning accuracy depending on this choice, as seen in Table 8.

---

> ### Author Rebuttal · Authors · 2026-03-30
>
> We sincerely thank the reviewer for the very positive feedback on our paper! Below, we try to address the listed concerns.
>
> **W1: The method relies on additional anchor datasets, requiring access to both harmful and alignment data, which may not always be available or well-matched in practice.**
>
> Thank you for the reviewer’s valuable suggestion. We acknowledge that Surgery relies on anchor datasets (i.e., harmful and alignment data) to compute sink divergence. But we tend to believe that this does not significantly limit its applicability in real-world scenarios.
>
> **(Evidence 1: Requires only a small amount of anchor datasets)**   In our experiments, only 1,000 harmful samples and 1,000 refusal samples are used.
>
> **(Evidence 2: There are opensourced harmful/resfusal dataset pair.)** While we acknowledge that Surgery require pairs of harmful /refusal data, which introduce some contraint empirically. However, we want to note that such constraint would not significantly hinder its applicabiliity because such pairs of data has been open-sourced and is made publicly available on Internet. For example, BeaverTails (https://github.com/domenicrosati/representation-noising/blob/main/data/beavertails_with_refusals_train.json) and LLM-LAT (https://huggingface.co/datasets/LLM-LAT/harmful-dataset) are two such datasets that provide pair of harmful/refusal data and are available to everyone.
>
>
> ---
>
> **W2: How sensitive is the method to the choice of the regularization coefficient $\lambda$, and how does this affect performance stability?**
>
> In this rebuttal, we further evaluate the sensitivity of \( \lambda \)  across models of different sizes and datasets. Please see the following results.
>
> **(With different model sizes $s$)**
> |  | **Harmful Score ↓** | **->** | **->** | **Finetune Accuracy ↑** | **->** | **->** |
> |--------|--------------------------|--------------------------|---------------------------|------------------------------|------------------------------|-------------------------------|
> | Methods | s=1B | s=8B | s=14B | s=1B | s=8B | s=14B |
> | SFT | 37.00 | 27.60 | 27.30 | **40.30** | 64.70 | 90.70 |
> | Surgery (λ = 100) | 24.40 | 9.00 | 12.80 | 38.90 | 68.50 | 90.50 |
> | Surgery (λ = 300) | **18.90** | **8.90** | **11.30** | 38.60 | **68.50** | **90.80** |
>
> **(Summary)** The results show that Surgery can provide consistent defense for different size of models **on the same hyper-parameter** $\lambda=300$. $\lambda=300$ achieves at least 16  reduction of harmful score, and at most 1.7 degradation of finetune accuracy accross different models.
>
> **(With different fine-tuning datasets)**
>
> |  |  Harmful Score ↓ | **->** | **->** | Finetune Accuracy ↑ | **->** | **->** |
> |--------|------------------------|--------------------------|---------------------------|-----------------------------|------------------------------|-------------------------------|
> | Methods | SST2 | GSM8K | AGNEWS | SST2 | GSM8K | AGNEWS |
> | SFT | 34.20 | 27.60 | 31.00 | 94.50 | 64.70 | **90.30** |
> | Surgery (λ=100) | 9.50 | 9.00 | 8.40 | 94.38 | 68.50 | 88.70 |
> | Surgery (λ=300) | **9.40** | **8.90** | **7.40** | **94.50** | **68.50**| 88.60 |
>
> **(Summary)** The results show that Surgery can provide consistent defense for different fine-tuning tasks **on the same hyper-parameter** $\lambda=300$, as $\lambda=300$ achieves at least 18.70 reduction of harmful score, and at most 1.7 degradation of finetune accuracy accross different tasks.
>
> Therefore, defenders can directly apply $\lambda=300$ to different sizes of model and different fine-tuning tasks without extensive tuning.
>
> ---
>
> **Q1: How does surgery differ from direct full-parameter gradient surgery?**
>
> In the revised version, we add a section in the appendix to discuss the differences between Surgery and direct full-parameter gradient surgery.
>
> **(Similarities)** Surgery and direct full-parameter gradient surgery (e.g., SPF/SafeGrad) share several commonalities: *(1) model safety degradation is attributed to intrinsic conflicts; (2) emphasis on optimizing the model’s internal mechanisms; (3) reliance on low-data-driven safety anchoring; and (4) dynamic online calibration during fine-tuning.*
>
> **(Differences)** Surgery differs from full-parameter gradient surgery in: (1) *optimization mechanism*: mechanism-based regularization vs. geometric projection; (2) *optimization granularity*: attention heads vs. fully parameter-level; (3) *hypothesis*: representation-level vs. gradient compatibility; and (4) *efficiency*: lightweight regularization vs. costly high-dimensional projections.
>
> **(Empirical analysis)** In the original version, we compared with SafeGrad. In the revised version, we plan to additionally include comparisons with SPF. We sincerely apologize that, due to the rebuttal deadline, these results could not be included; we will provide the experimental results against SPF during the discussion phase.

---

> > ### Author Rebuttal · Reviewer_dUh7 · 2026-04-03
> >
> > The rebuttal have addressed my concerns.

---

> > > ### Author Response · Authors · 2026-04-03
> > >
> > > We are pleased to have addressed the reviewers’ concerns, and we sincerely thank you for your valuable suggestions, which have helped improve and strengthen this work.
> > >
> > > Below, we present experimental comparisons with direct full-parameter gradient surgery methods (SPF and SafeGrad). In the camera ready, we will add SPF as baseline comparison in our Table 1-8 given that SPF and SafeGrad represents an important  line of defenses (i.e., gradient projection) against fine-tuning attack.
> > >
> > > |Methods|HS↓|FA↑|
> > > |-|-|-|
> > > |SFT|27.60|64.70|
> > > |SafeGrad|18.5|65.30|
> > > |SPF|21.60|68.40|
> > > |Surgery|8.90|68.50|

---

### Official Review · Reviewer_cWSM · 2026-03-09

**Soundness:** 3
**Presentation:** 3
**Significance:** 3
**Originality:** 3
**Overall Recommendation:** 4
**Confidence:** 4

**Summary:**

This paper first introduces sink divergence to measure the difference in attention sink values between harmful and refusal datasets. Based on the observation that attention heads with positive sink divergence correlate with increased model harmfulness, this paper proposes Surgery, a regularizer for suppressing positive sink divergence during fine-tuning, steering them toward the negative group to mitigate harmful tuning. Experimental results across various models and benchmarks show that Surgery significantly reduces harmful scores while maintaining utility performance on downstream tasks.

**Compliance With Llm Reviewing Policy:**

Affirmed.

**Final Justification:**

The authors conducted very detailed supplementary experiments during the rebuttal. My concerns have been adequately addressed.

**Key Questions For Authors:**

**Q1**: This paper mainly targets the FaaS scenario. However, in practice, when users attempt to fine-tune commercial LLMs (e.g., GPT-4o) through API-based services using harmful datasets, the process will be blocked by the service provider. A more realistic harmful tuning scenario is using only benign data to destroy the model's safety alignment [1–4]. It would be valuable to discuss whether Surgery remains effective under this setting.

**Q2**: The defense performance of Surgery relies heavily on the selection of anchor datasets ($X_m$ and $X_r$). As noted in [5], harmful data from different distributions may exhibit substantially different characteristics. If the anchor data and the evaluation data come from different distributions, it remains unclear whether Surgery would still be effective. It would be helpful for the authors to discuss the robustness of the method under such distribution shifts, or provide empirical results evaluating this scenario.

**Q3**: The utility evaluation in this paper appears to be conducted in-distribution. For instance, on GSM8K, there is no significant performance difference between the fine-tuned model and the base model. It would be better to examine the method under a more out-of-distribution fine-tuning setting. For example, the authors could fine-tune the model on the full Alpaca dataset using Surgery, and then evaluate both the model's defense against harmful queries and its utility on GSM8K.

[1] He, Luxi, Mengzhou Xia, and Peter Henderson. "What is in your safe data? identifying benign data that breaks safety." arXiv preprint arXiv:2404.01099 (2024).

[2] Zhao, Wei, Zhe Li, Yige Li, and Jun Sun. "Unleashing the Unseen: Harnessing Benign Datasets for Jailbreaking Large Language Models." arXiv preprint arXiv:2410.00451 (2024).

[3] Guan, Zihan, Mengxuan Hu, Ronghang Zhu, Sheng Li, and Anil Vullikanti. "Benign samples matter! fine-tuning on outlier benign samples severely breaks safety." arXiv preprint arXiv:2505.06843 (2025).

[4] Xie, Zhixin, Xurui Song, and Jun Luo. "Attack via Overfitting: 10-shot Benign Fine-tuning to Jailbreak LLMs." arXiv preprint arXiv:2510.02833 (2025).

[5] Wang, Cheng, Zeming Wei, Qin Liu, and Muhao Chen. "False sense of security: Why probing-based malicious input detection fails to generalize." arXiv preprint arXiv:2509.03888 (2025).

**Limitations:**

Yes

**Strengths And Weaknesses:**

#### **Strengths**
* Unlike previous works which focus on hidden states in the residual stream, this paper relates the role of attention sinks with harmful data, providing a new perspective for understanding LLMs safety mechanisms.
* Surgery exhibits computational efficiency compared to existing baselines, and shows both strong defense performance against harmful tuning while maintaining model utility.

---

#### **Weaknesses**
Major concerns
* Surgery appears to be highly sensitive to hyperparameters, which may make it difficult to adapt efficiently to models of different sizes or to new datasets without extensive tuning.
* Surgery adopts a high regularizer intensity ($\lambda$=300), which raises significant concerns about potential over-refusal problem. It would be better to evaluate the fine-tuned model on over-refusal benchmarks like OKTest (https://arxiv.org/abs/2401.17633) and FalseReject (https://arxiv.org/abs/2505.08054).
* The experimental scale is somewhat limited. With only 1k training examples, full-parameter fine-tuning may lead to overfitting or catastrophic forgetting. A more practical setup should involve PEFT methods (e.g., LoRA) for experiments with 1k data points, while full-parameter fine-tuning would be more appropriate for larger datasets (e.g., 100k examples).

Minor concerns
* Typo: line 206 Figure reffig:motivate $\rightarrow$ Figure 3

---

> ### Author Rebuttal · Authors · 2026-03-30
>
> Thank the reviewer for the valuable suggestions.
>
> **W1: Evaluate the sensitivity of λ across different model sizes and datasets.**
>
> **(With different model sizes)**
>
> | |HS↓|->|->|FA↑|->|->|
> |-|-|-|-|-|-|-|
> |Methods|s=1B|s=8B|s=14B|s=1B|s=8B|s=14B|
> |SFT|37.0|27.60|27.3|**40.3**|64.7|90.7|
> |Surgery(λ=100)|24.4|9.0|12.8|38.9|68.5|90.5|
> |Surgery(λ=300)|**18.9**|**8.9**|**11.3**|38.6|**68.5**|**90.8**|
> ---
>
> **(Summary)** The results show that Surgery can provide consistent defense for different size of models **on the same hyper-parameter** $\lambda=300$. $\lambda=300$ achieves at least 16 reduction of harmful score, and at most 1.7 degradation of finetune accuracy across different models.
>
> **(With different datasets)**
>
> | |HS↓|->|->|FA↑|->|->|
> |-|-|-|-|-|-|-|
> |Methods|SST2|GSM8K|AGNEWS|SST2|GSM8K|AGNEWS|
> |SFT|34.2|27.6|31.0|94.5|64.7|**90.3**|
> |Surgery(λ=100)|9.5|9.0|8.4|94.38|68.5|88.7|
> |Surgery(λ=300)|**9.4**|**8.9**|**7.4**|**94.5**|**68.5**|88.6|
>
> **(Summary)** The results show that Surgery can provide consistent defense for different fine-tuning tasks **on the same hyper-parameter** $\lambda=300$, as $\lambda=300$ achieves at least 18.70 reduction of harmful score, and at most 1.7 degradation of finetune accuracy across different tasks.
>
> Therefore, defenders can directly apply $\lambda=300$ to different sizes of model and different fine-tuning tasks without extensive tuning.
>
> **W2: Does Surgery with λ = 300, lead to over-refusal?**
>
> **(Why a relatively large λ is required).** The sink divergence is typically very small (mostly within [-0.05, 0.05]), thus, a relatively large λ is needed.
>
> Following FalseReject, we evaluate the model’s $USR_{benign}$ (shortened with USR) on both OKTest and FalseReject. Higher USR means that model is less over-refusal.
>
> |Methods|HS↓|USR↑|->|
> |-|-|-|-|
> |Methods||OKTest|FalseReject|
> |SFT|27.6|**94.22**|**78.2**|
> |DSS|16.8|86.68|69.7|
> |SPARD|16.8|87.67|72.22|
> |ConstrainedSFT|17.6|84.6|68.4|
> |Surgery|**8.9**|89.0|72.8|
>
> Visualization examples: https://anonymous.4open.science/r/Surgery-A69E/rebuttal/over-refusal_sample.pdf
>
> **(Summary)** Surgery consistently achieves higher USR than other defenses (e.g., on FalseReject, Surgery improves USR over ConstrainedSFT by 4.40% ), demonstrating that it introduces less over-refusal while maintaining strong safety performance.
>
> **W3: Evaluation under the LoRA.**
>
> We conduct additional experiments with LoRA.
>
> |Methods|HS↓|FA↑|
> |-|-|-|
> |SFT|34.4|56.2|
> |SPARD|40.4|59.1|
> |DSS|19.8|**63.0**|
> |Surgery|**15.7**|61.3|
>
> **(Result)** Surgery achieves better defensive performance under the LoRA setting compared with the baselines.
>
> **W4: Correct typos.**
>
> Thanks for pointing out. In the revision, we will correct these typos.
>
> ---
>
> **Q1: Evaluated under benign data attack.**
>
> We next evaluate different defenses with clean fine-tuning data as well as the mentioned benign data attack [1][2][3][4].
>
> | |HS↓|->|->|->|->|FA↑|->|->|->|->|
> |-|-|-|-|-|-|-|-|-|-|-|
> |Methods|clean|[1]|[2]|[3]|[4]|clean|[1]|[2]|[3]|[4]|
> |SFT|15.1|17.3|19.1|16.8|14.9|67.2|64.4|67.8|65.5|64.5|
> |DSS|13.5|13.6|16.1|14.6|14.8|59.8|**67.6**|55.2|57.9|51.6|
> |SPARD|12.4|14.6|16.6|13.8|14.1|67.9|66.2|67.8|68.5|67.8|
> |Surgery|**5.5**|**6.8**|**5.9**|**5.4**|**8.1**|**69.4**|67.4|**67.8**|**69.8**|**69.0**|
>
> **(Observation 1)** With clean fine-tuning data,  Surgery achieves the best defensive and inference performance (5.50% and 69.40%).
>
> **(Observation 2)** Under the four benign data attack settings, baselines exhibit degraded defensive performance (e.g., SFT’s HS increases by 4% under [2]), while Surgery shows minimal fluctuation, with HS remaining around 6%.
>
> **Q2: Evaluate under distribution shifts between anchor and harmful datasets.**
>
> We test two different datasets Beavertails and RepNoise as anchor datasets, and three different datasets as harmful dataset.
>
> |Methods|Anchor Dataset|HS(BeaverTails)↓|HS(Harmbench)↓|HS(Sorrybench)↓|
> |-|-|-|-|-|
> |SFT|—|27.6|28.75|30.91|
> |SPARD|—|20.6|30.0|27.73|
> |DSS|—|16.8|24.5|25.68|
> |Surgery|BeaverTails|**8.9**|**9.5**|**12.95**|
> |Surgery|RepNoise|13.2|12.4|16.36|
>
> **(Summary)** Compared with the baselines, under both anchor datasets, Surgery achieves the best defensive performance across three harmful datasets with different distributions.
>
> **Q3: Evaluation under out-of-distribution (OOD) fine-tuning.**
>
> We test the model performance after fine-tuning (FT) on Alpaca and evaluate on GSM8K. The results are as follows:
>
> |Methods|HS(before FT)↓|HS(after FT)↓|FA(before FT )↑|FA(after FT )↑|
> |-|-|-|-|-|
> |SFT|14.8|30.8|50.2|**56.7**|
> |Surgery|14.8|**16.3**|50.2|54.0|
>
> **(Summary)** The results show that under OOD fine-tuning (Alpaca → GSM8K), both SFT and Surgery improve fine-tuning accuracy, but SFT incurs a much larger increase in harmful score (from 14.80% to 30.80%) compared to Surgery (from 14.80% to 16.30%). **This indicates that Surgery achieves better defense performance while maintaining reasoning performance under OOD fine-tuning.**

---

> > ### Author Rebuttal · Reviewer_cWSM · 2026-04-03
> >
> > Thank you for the detailed clarification. It would be beneficial to incorporate these points into the main paper, especially the experiments on over-refusal evaluation and attacks using benign data. I will raise my score to 4.

---

> > > ### Author Response · Authors · 2026-04-04
> > >
> > > Thank you for the positive acknowledgment of our work!
> > >
> > > We sincerely appreciate your insightful questions and valuable suggestions. In the revision, we will incorporate these points into the main paper, which will further improve and strengthen the paper. Please feel free to share any further thoughts during the discussion phase with other reviewers.
> > >
> > > Thanks,
> > >
> > > Authors

---

### Official Review · Reviewer_mfPz · 2026-03-12

**Soundness:** 3
**Presentation:** 4
**Significance:** 3
**Originality:** 3
**Overall Recommendation:** 5
**Confidence:** 4

**Summary:**

This paper tries to solve the harmful fine-tuning by a fine-tuning-stage defense. The authors find that attention heads tend to shift toward the positive sink divergence group. Based on the finding, they propose Surgery, which employs sink divergence suppression regularizer during the fine-tuning
stage to steer attention heads toward the negative sink divergence group. Comprehensive results show the superiority of their method.

**Compliance With Llm Reviewing Policy:**

Affirmed.

**Key Questions For Authors:**

see weakness

**Limitations:**

yes

**Strengths And Weaknesses:**

### Strength

- The finding of sink divergence is very interesting, and the analysis experiments make the observation convincing.
- The proposed method is well motivated by primary studies, which makes it easy to understand.
- The paper is well written, and the experimental results are well presented, making it easy to follow.
- The evaluation is extensive, including multiple models, datasets, fine-tuning settings, and ablation and efficiency analyses.

### weakness

- The main weakness is the theoretical foundation. All hypothesis is empirically verified but not theoretically established. Some theories may give insights into understanding this finding.
- Another concern is about applying to different sink mechanisms like GPT-OSS.

---

> ### Author Rebuttal · Authors · 2026-03-30
>
> We sincerely thank the reviewer for the very positive feedback on our paper! Below, we try to address the listed concerns.
>
> **W1: Lack of theoretical analysis.**
>
> We thank the reviewer pointing out this weakness.  We acknowledge that exploring the theoretical implication of sink divergence is important and particularly of interest.  However, we would like to leave this exploration a future work. The paper itself, as the reviewer correctly point out, is based on empirical experiments, and we sincerely hope that the lack of theoretical foundation will not significantly vitiate the value of this paper, considering our strenghth in empricial analysis. To reflect the reviewer's concern, we have updated our limitation section to include "lack of theoretical alsysis" as one primary limitation.
>
> ---
>
> **W2: Surgery can be readily generalized to different sink mechanisms like GPT-OSS?**
>
> We sincerely thank the reviewer for the valuable suggestion. In the revision, we conduct experiments on the GPT-OSS-20B model, fine-tuned on GSM8K.  Please see the following results.
>
> **(Observation 1: GPT-OSS-20B exhibits a pronounced attention sink phenomenon.)** As illustrated in the figure, we visualize the attention patterns of the 10th head in layer 7 and the 20th head in layer 22 of the GPT-OSS-20B model. We observe that GPT-OSS-20B exhibits a clear attention sink phenomenon, with certain tokens (e.g., the 16th token) consistently attracting disproportionately close attention. This observation indicates that the Surgery applies to GPT-OSS.
>
> https://i.postimg.cc/N9ZFXNGW/attention-patterns-7-10.png
>
> https://i.postimg.cc/47jyc8NC/attention-patterns-22-20.png
>
> **(Performance of GPT-OSS-20B)**
>
> | Methods | Harmful Score ↓ | Finetune Accuracy ↑ |
> |--------|------------------|----------------------|
> | SFT | 34.70 | 74.00 |
> | Surgery | **14.40** | **78.60** |
>
> **(Observation 2: Surgery achieves competitive defensive performance on GPT-OSS-20B.)** The above results show that Surgery improves defensive performance and reasoning performance by 20.30% and 4.60%, respectively, compared to SFT. These results demonstrate that Surgery can be effectively transferred to GPT-OSS-20B, further indicating its generalizability to different sink mechanisms.
>
> **(Result)** First, GPT-OSS-20B exhibits a pronounced attention sink phenomenon. Then, experimental results show that Surgery achieves competitive defensive performance on GPT-OSS-20B. Thus, **Surgery can generalize to different sink mechanisms.**

---

> > ### Author Rebuttal · Reviewer_mfPz · 2026-04-02
> >
> > Thank you for the author’s feedback. I will keep my score and vote to accept.

---

> > > ### Author Response · Authors · 2026-04-02
> > >
> > > Thank you for the positive acknowledgment of our work! Please never hesitate to reach out if you have more thoughts during our interaction with other reviewers.
> > >
> > > Thanks,
> > >
> > > Authors

---

### Official Review · Reviewer_ZkFC · 2026-03-13

**Soundness:** 3
**Presentation:** 3
**Significance:** 3
**Originality:** 3
**Overall Recommendation:** 5
**Confidence:** 3

**Summary:**

The authors analyze the role of the attention sink mechanism during fine-tuning and introduce a statistic called sink divergence to characterize the behavior of individual attention heads. And author based on this observation propose a defense method Surgery by suppressing the sink divergence.

**Compliance With Llm Reviewing Policy:**

Affirmed.

**Key Questions For Authors:**

Please see above

**Limitations:**

yes

**Strengths And Weaknesses:**

**Strength**

1. Well-written paper.
The paper is clearly written and generally easy to follow. The method and experimental setup are described in a structured and understandable way.

2. Novel perspective.
The use of the attention sink mechanism is interesting and relatively under-explored in the context of safety alignment. Studying attention-level signals to diagnose harmful behavior is a novel angle.

3. Insightful preliminary analysis.
The preliminary experiments exploring the relationship between attention sink divergence and model harmfulness are very interesting. The analysis provides empirical evidence suggesting that certain attention head behaviors correlate with safety degradation during harmful fine-tuning.

4. Extensive evaluation.
The paper evaluates the proposed method on multiple safety benchmarks, and the reported improvements are relatively strong. The experimental results support the effectiveness of the proposed defense method.


**weakness**
1. Dependence on harmful/refusal samples for computing sink divergence.

It is unclear whether the harmful and refusal samples used to compute attention sink are the same samples used to poison the fine-tuning dataset. If they overlap, the experimental setup may implicitly assume access to information that may not be available in realistic scenarios. In practice, identifying harmful or refusal samples within large fine-tuning datasets can be challenging. It would be helpful if the authors could clarify:

whether these datasets overlap,

how the method performs when the harmful/refusal samples differ from the poisoned data, and

whether there are strategies for estimating sink divergence without explicit access to labeled harmful or refusal samples. And would it affect the performance?

2. Motivation linking safety and attention sink behavior.

While the empirical observations are interesting, the paper does not clearly explain why attention sink behavior should be related to safety alignment. The motivation for associating sink divergence with harmful pattern learning could be strengthened. Providing additional intuition, theoretical reasoning, or mechanistic analysis would make the proposed hypothesis more convincing.

---

> ### Author Rebuttal · Authors · 2026-03-30
>
> We sincerely thank the reviewer for the very positive feedback! Below, we try to address the listed concerns.
>
> **W1: Do the harmful/refusal samples used for computing sink divergence overlap with the poisoned fine-tuning data? How robust is the method when these samples differ, and can sink divergence be estimated without explicit harmful/refusal annotations?**
>
> **(① The harmful and refusal samples used to compute sink divergence do not overlap with the poisoned harmful data.)**  Specifically, the samples used for computing sink divergence are drawn from the training split of the *BeaverTails* dataset (indices 0–1000), while the poison samples are sourced from *RepNoise-Refusal* (starting from index 5000).
>
> **(② The distribution shift between anchor datasets and evaluation harmful datasets has minimal impact on performance.)**  Originally, we used BeaverTails as the anchor dataset and evaluated poisoned data from BeaverTails, Harmbench, and Sorrybench; in the revision, we additionally introduced RepNoise-Refusal as an anchor dataset while keeping the same evaluation setup. Please see the following results.
>
> **(Performance of Surgery under distribution shift.)**
> | Methods | Anchor Dataset | HS (BeaverTails) ↓ | HS (Harmbench) ↓ | HS (Sorrybench) ↓ |
> |-|-|-|-|-|
> | SFT | — | 27.6 | 28.75 | 30.91 |
> | Lisa | — | 14.8 | 20.75 | 22.73 |
> | SafeGrad | — | 18.5 | 23.5 | 23.64 |
> | ConstrainedSFT | — | 17.6 | 21.25 | 28.41 |
> | AsFT | — | 16.1 | 22.50 | 22.5 |
> | SPARD | — | 20.6 | 30.0 | 27.73 |
> | DSS | — | 16.8 | 24.5 | 25.68 |
> | Surgery (BeaverTails) | BeaverTails | **8.9** | **9.5** | **12.95** |
> | Surgery (RepNoise-Refusal) | RepNoise-Refusal | 13.2 | 12.4 | 16.36 |
>
> **(Result)** The results show that Surgery achieves the best defense performance across all three harmful datasets under both anchor datasets, indicating that it remains robust even under distribution shifts between anchor and evaluation datasets.
>
> **(③ Sink divergence is currently estimated in a sample-based manner.)** Sink divergence reflects the model’s attention sink behavior on harmful and refusal samples. Therefore, this metric is primarily evaluated through sample-based estimation.
>
> **(Summary)** Overall, while collecting a small amount of relevant anchor datasets can further enhance the performance of Surgery, competitive defensive performance can still be achieved even in data-constrained scenarios by leveraging publicly available datasets as substitutes.
>
> ---
>
> **W2: What is the underlying motivation for linking attention sink behavior to safety alignment, and why should sink divergence correlate with harmful pattern learning? Providing additional intuition, theoretical reasoning, or mechanistic analysis.**
>
> **(Empirical analysis)** In Appendix E, we empirically investigate the relationship between sink divergence and the *safety vector*. Specifically, we define the negative direction of the safety vector as the *harmful vector*, and, inspired by this notion, introduce a statistical metric termed *harmful divergence*.
>
> **(Observation 1)** As shown in the middle of Figure 9, we observe that, **similar to sink divergence, harmful divergence separates attention heads into two distinct groups**.
>
> **(Observation 2)** As shown on the right of Figure 9, we observe that **some attention heads are assigned to different groups by harmful divergence and sink divergence**.
>
> **(Observation 3)** As shown in Table 9, **suppression based on harmful divergence is suboptimal**.
>
> **(Result of empirical analysis)** *Sink divergence and the safety vector share certain similarities in that both quantify harmful representations. However, they operate on different statistical objects*: the safety vector (and harmful divergence) captures variations in hidden activation magnitudes, whereas sink divergence directly measures attention sink behavior. As attention sinks more directly govern information routing during inference, sink divergence better captures the functional mechanisms underlying harmful generation, leading to more effective mitigation.
>
> **(Theoretical Analysis)** In the revision, we provide a new theoretical result showing that **sink divergence is proportional to the scalar projection of the harmful vector (the negative direction of the safety vector)**, as illustrated below.
>
> **Theorem.**
> The sink divergence $d_h$ of head $h$ is proportional to the scalar projection of the harmful vector $\mathbf{v}_h$ onto the sink subspace:
>
> $d_h \approx (1 / ||v_{sink}||) Proj_{v_{sink}}(v_h)$
>
> **Proof.** Due to the rebuttal word limit, please refer to https://anonymous.4open.science/r/Surgery-A69E/rebuttal/proof.pdf for the detailed proof.
>
> **(Summary)** Through theoretical analysis, we show that the sink divergence is proportional to the scalar projection of the harmful vector onto the sink subspace. Furthermore, empirical results demonstrate that, while sink divergence shares similarities with harmful divergence, it consistently outperforms the latter.

---

> > ### Author Rebuttal · Reviewer_ZkFC · 2026-04-03
> >
> > Thanks so much for the rebuttal, my concerns are addressed, so I will raise my score to 5.

---

> > > ### Author Response · Authors · 2026-04-03
> > >
> > > Thank you for the positive acknowledgment of our work!  We are pleased to have addressed the reviewers’ concerns, and we sincerely appreciate your valuable suggestions, which have helped improve and strengthen the paper. Please feel free to share any further thoughts during the discussion phase with other reviewers.
> > >
> > > Best regards,
> > >
> > > The Authors

---

### Decision · Program_Chairs · 2026-04-30

**Decision:**

Accept (spotlight)

**Comment:**

Thanks for your submission to ICML 2026. The submission identifies a novel and interesting phenomenon: number of positive sink divergence increases along with the increase of model's harmfulness. Based on the finding, the submission proposes a finetuning-stage defense named Surgery. Surgery utilizes a regularizer for sink divergence suppression and achieves superior performance.

All reviewers appreciate the novelty of the finding and its strong practical implications in terms of providing the effective, low-cost, and generalizable finetuning-stage defense. During the rebuttal phase, all reviewers were active involved and the authors provided extensive additional evidences to address all reviewers' concerns. As the result, all reviewers unanimously vote for acceptance. Please make sure to incorporate these additional results in the camera-ready version (including the connection to harmful vectors; experiments in OOD harmful/refusal samples, hyperparameter sensitivity, different sink mechanisms, over-refusal, benign data attacks, PEFT, and the gradient survery baselines).

In addition, if space permits, I would personally happy to see a discussion of the applicability in LLM architectures that mitigate attention sinks such as Gated Attention [1].

[1] Qiu, Z., Wang, Z., Zheng, B., Huang, Z., Wen, K., Yang, S., ... & Lin, J. Gated Attention for Large Language Models: Non-linearity, Sparsity, and Attention-Sink-Free. NeurIPS 2025.